# Effects of enhanced downwelling of $NO_x$ on Antarctic upper stratospheric ozone in the 21st century

Ville Maliniemi[1], Hilde Nesse Tyssøy[1], Christine Smith-Johnsen[1], Pavle Arsenovic[2], and Daniel R. Marsh[3,4]

[1]Birkeland Centre for Space Science, Department of Physics and Technology, University of Bergen, Norway
[2]EMPA Swiss Federal Laboratories for Material Science and Technology, Zurich, Switzerland
[3]National Center for Atmospheric Research, Boulder, CO, USA
[4]Faculty of Engineering and Physical Sciences, University of Leeds, UK

**Correspondence:** Ville Maliniemi (ville.maliniemi@uib.no)

**Abstract.** Ozone is expected to fully recover from the CFC-era by the end of the 21st century. Furthermore, because of anthropogenic climate change, a cooler stratosphere accelerates ozone production and is projected to lead to a super recovery of ozone. We investigate the ozone distribution over the 21st century with four different future scenarios using simulations of the Whole Atmosphere Community Climate Model (WACCM). At the end of the 21st century, equatorial upper stratosphere has roughly 0.5 to 1.0 parts per million more ozone in scenario with the highest greenhouse gas emissions compared to conservative scenario. Polar ozone levels exceed those in the pre CFC-era in scenarios that have the highest greenhouse gas emissions. This is true in the Arctic stratosphere and the Antarctic lower stratosphere. The Antarctic upper stratosphere is an exception, where different scenarios all have similar levels of ozone during winter, which do not exceed pre-CFC levels. Our results show that this is due to excess nitrogen oxides ($NO_x$) descending faster from above in the stronger scenarios of greenhouse gas emissions. $NO_x$ in the polar thermosphere and upper mesosphere is mainly produced by energetic electron precipitation (EEP) and partly by solar UV via transport from low latitudes. Our results indicate that the thermospheric/upper mesospheric $NO_x$ will be important factor for the future Antarctic ozone evolution, and could potentially prevent a super recovery of ozone in the upper stratosphere.

## 1 Introduction

Stratospheric ozone experienced a dramatic decrease from the 1960s until the 1990s due to anthropogenic chloro-fluoro-carbon (CFC) emissions (Cicerone, 1987; Anderson et al., 1991) and the associated increase of reactive chlorine oxides ($ClO_x$) in the stratosphere. Since then, the Montreal Protocol has been able to limit the use of CFCs (Velders et al., 2007), and in the beginning of the 21st century stratospheric ozone is showing signs of recovery (Solomon et al., 2016).

Greenhouse gas emissions also alter the stratospheric ozone (Langematz, 2018). As a consequence of higher levels of carbon dioxide ($CO_2$), the stratosphere is cooling, which increases the rate of stratospheric ozone production (Li et al., 2009). This is projected to lead to a super recovery of ozone, i.e., higher ozone concentrations than before the 1960s, especially in the upper stratosphere (WMO, 2018, Chapter 4). In addition to the impact on chemical ozone production, climate change is modulating

ozone through changes in the atmospheric circulation. Climate models predict that the Brewer-Dobson circulation (BDC) is increasing (Garcia and Randel, 2008; Butchart, 2014) which leads to enhanced transport of ozone into the polar lower stratosphere, and a reduction of ozone in the equatorial lower stratosphere (Langematz, 2018; Shepherd, 2008). This transport effect is projected to be stronger in the northern hemisphere, leading to a more prominent ozone super recovery in the Arctic lower stratosphere than in the Antarctic lower stratosphere (WMO, 2018, Chapter 4).

Polar stratospheric ozone is also impacted from above. Energetic electron precipitation (EEP) from the magnetosphere produces reactive nitrogen oxides ($NO_x$) in the thermosphere and upper mesosphere (Andersson et al., 2018). Sporadic solar proton events can also produce $NO_x$ in the polar mesosphere (Jackman et al., 2008). Solar UV absorbed in the lower thermosphere also produces $NO_x$, which can be transported to polar latitudes (Gérard et al., 1984). The chemical lifetime of $NO_x$ in the mesosphere and lower thermosphere is enhanced during wintertime polar darkness due to the absence of photolysis. This allows $NO_x$ to be transported to the upper stratosphere with the prevailing vertical residual circulation (Randall et al., 2006; Funke et al., 2014), where it depletes ozone in a catalytic reaction (Lary, 1997). Recently, Maliniemi et al. (2020) showed in the Whole Atmosphere Community Climate Model (WACCM) that this stratospheric indirect $NO_x$ will increase substantially during the 21st century in the southern hemisphere in scenarios with increasing greenhouse gas emissions. Similar results have been obtained earlier with the EMAC chemistry-climate model (Baumgaertner et al., 2010). This is a consequence of stronger mesospheric descent in the future Antarctic, while no such strengthening of the mesospheric descent is predicted in the Arctic (Maliniemi et al., 2020).

In this paper we investigate the ozone distribution over the 21st century under four different future scenarios using the WACCM chemistry-climate model. We concentrate on polar stratospheric variability during winter, but also show results over the whole middle atmosphere and during all seasons. Section 2 describes the data and statistical methods. Section 3 provides results divided to three subsections: the polar winter ozone evolution from the pre-industrial era until the end of the 21st century; differences in the global ozone distribution at the end of the 21st century between the strongest and conservative future scenarios regarding their greenhouse gas emissions; and same for the polar ozone. A summary is given in Section 4.

## 2   Data and Methods

The data used in this study are from simulations of a free-running version of WACCM6 within CESM2. The model components and parametrisations are described in detail by Marsh et al. (2013), with updates detailed by Gettelman et al. (2019). Five different simulations are analyzed. Historical simulations (3 ensemble members) cover the period 1850-2014 (Coupled Model Intercomparison Project 6 (CMIP6) DECK simulations). Four different future scenario (CMIP6 ScenarioMIP: SSP1, SSP2, SSP3 and SSP5 (Shared Socioeconomic Pathway)) simulations cover the period 2015-2100 (O'Neill et al., 2016). SSP1 and SSP3 have 1 ensemble member and SSP2 and SSP5 have 5 ensemble members. For the historical, SSP2 and SSP5 model simulations, the results shown here are ensemble means.

Different SSPs include a wide range of future actions by society, including greenhouse gas emissions. Global average $CO_2$ concentrations in 2100 are 446 parts per million (ppm) in SSP1, 603 ppm in SSP2, 867 ppm in SSP3 and 1135 ppm in SSP5

(Meinshausen et al., 2020). The radiative forcing increase of the climate system by 2100 relative to pre-industrial era is 5.0 W/m$^2$ in SSP1, 6.5 W/m$^2$ in SSP2, 7.2 W/m$^2$ in SSP3 and 8.7 W/m$^2$ in SSP5. The details of the different SSPs can be obtained from Riahi et al. (2017). All model runs are forced with solar activity following the recommendations of CMIP6. This provides estimates of the solar activity before the space era, and a future solar forcing scenario (Matthes et al., 2017). Solar forcing consists of total and spectral solar irradiance, as well as galactic cosmic rays, solar proton events and energetic electron precipitation. All SSPs have the same future solar activity scenario (called the "reference scenario," see details in Matthes et al. (2017)).

We concentrate on monthly mean zonal mean volume mixing ratios of ozone, NO$_x$ and ClO$_x$, as well as temperature and zonal wind. The latitudinal resolution of the model is 0.94° (192 bins) and altitude ranges from the surface up to ≈140 km (in 70 levels). In this study we focus on altitudes from around the mesopause to the surface (0.01 hPa to 1000 hPa). We analyse the centennial time series (1850-2100) of ozone and ClO$_x$ concentrations in the polar stratosphere, as well as the Brewer-Dobson circulation in the equatorial stratosphere. The smooth long-term variations shown in Figures 1, 2, 3 and 7 are calculated using the LOWESS-method (LOcally WEighted Scatterplot Smoothing) applied with a 31-year window (Cleveland and Devlin, 1988). More details of the method can be found in (Maliniemi et al., 2014). The relative change in ozone and ClO$_x$ over the whole atmosphere from 1960 to 2000 shown in Figure 4 was calculated by subtracting a 5-year mean centered on 1960 from a 5-year mean centered on 2000. Significance was calculated using a Mann-Kendall test (Mann, 1945).

We subtract SSP2 means from those of SSP5 to evaluate the differences in ozone (Figure 5 and 9), NO$_x$ (Figure 8 and 9), temperature and zonal wind (Figure 6) during 2090-2100 period, i.e., SSP5[ensemble mean]-SSP2[ensemble mean]. Statistical significance for the differences between SSP5 and SSP2 during 2090-2100 are calculated applying a Monte Carlo method: we take a random 11-year time period from 2015-2100 and calculate the difference in each latitude/height bin. This is performed 1000 times and original value (difference in 2090-2100) is compared to the distribution of these 1000 repetitions to obtain fraction of more extreme differences (both tails of the distribution). This fraction then represents the p-value in each bin with the null hypothesis that there is no difference between SSP5 and SSP2.

In addition, we use a method proposed by Wilks (2016) called a false detection rate. This is done because our results for 2090-2100 differences (and relative change from 1960 to 2000 in Figure 4) are presented over several latitudes and altitudes, and thus have a multiple hypothesis testing situation. This method adjusts the p-values to take into account the spatial autocorrelation and the fact that the probability of erroneously rejecting the null hypothesis increases with the number of individual hypothesis tests. Thus, after the procedure, we obtain a global significance of 95% of the whole presented grid, which means that the probability of erroneously rejecting the (individual) null hypothesis will be 5%.

# 3   Results

## 3.1   Centennial polar winter ozone in different future scenarios

Figure 1 shows the late winter polar total column ozone time series for both hemispheres. The minimum level of ozone is reached a few years after 2000 (Solomon et al., 2016). Ozone returns to 1980s level around 2050 in the southern hemisphere

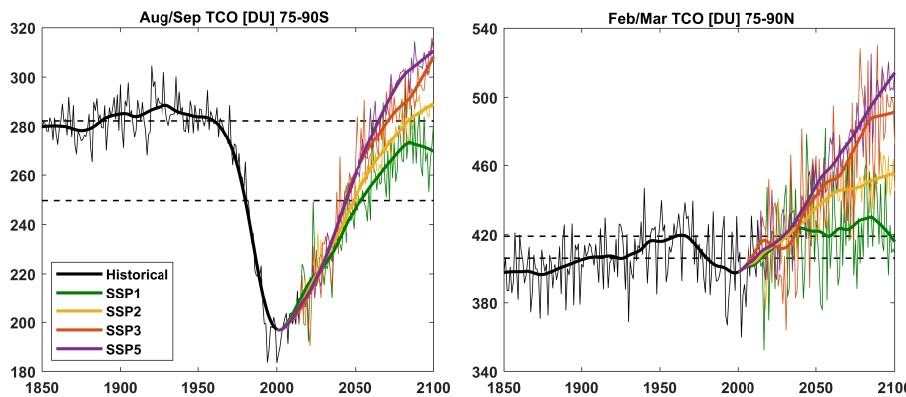

**Figure 1.** The time series of Aug/Sep Antarctic (left) and Feb/Mar Arctic (right) total column ozone from 1850 to 2100 in Dobson units. Black=1850–2014 historical run (mean of 3 ensemble members), green=2015–2100 SSP1 (one simulation), yellow=2015–2100 SSP2 (mean of 5 ensemble members), red=2015–2100 SSP3 (one simulation), and purple=2015–2100 SSP5 (mean of 5 ensemble members). Thin lines are yearly averages and thick lines are the 31-year smoothed trend calculated with LOWESS-method. Smoothed trends for SSPs are calculated continuously with the historical run to avoid gaps around 2015 (note the SSP thick lines starting from year 2000). Dotted lines represent the average level for years 1960 and 1980.

and a little bit earlier in the northern hemisphere. Columns in the different future scenarios begin to diverge from each other after 2050. Both polar regions show a super recovery in SSP3 and SSP5, i.e., the column ozone exceeds 1960 levels towards the end of the 21st century, which is more notable in the northern hemisphere. One can also see that yearly variability (thin lines in Figure 1) is somewhat larger in SSP1 and SSP3. This is because there is just one ensemble member for those SSPs, while the SSP2 and SSP5 results are the mean of 5 ensemble members.

Figure 2 shows the time series of late winter $ClO_x$ in the Antarctic stratosphere. One can see that the maximum level of $ClO_x$ coincides with the minimum in ozone around year 2000. After that, $ClO_x$ starts to decrease as a result of the Montreal protocol (Velders et al., 2007). All different future scenarios have approximately the same evolution of stratospheric $ClO_x$, due to all SSPs having the same WMO future scenario for CFCs (Meinshausen et al., 2020). We note that in the Arctic stratosphere the evolution of $ClO_x$ across the various SSPs is also very similar (not shown).

Figure 3 shows the evolution of the mean ozone volume mixing ratio for the different SSPs in the upper and the lower stratosphere in both polar regions. Lower stratospheric ozone changes are very similar to those of total column ozone in both the Antarctic and the Arctic as would be expected since the majority of the ozone is in the lower stratosphere. Upper and lower stratospheric ozone in the Arctic shows a super recovery in SSP3 and SSP5, and the decrease from 1960 to 2000 is notably larger in the upper stratosphere. In the Antarctic upper stratosphere ozone only returns to roughly 1960 level in all SSPs; no super recovery is predicted in any of the SSPs and the level of ozone in SSP5 is slightly less than in SSP2 and SSP3 at the end of the 21st century. Sudden decreases of ozone in yearly values (downward spikes in thin lines) can also be seen in the Antarctic upper stratosphere. These have been previously shown to be due to the large solar proton events (SPE) during winter

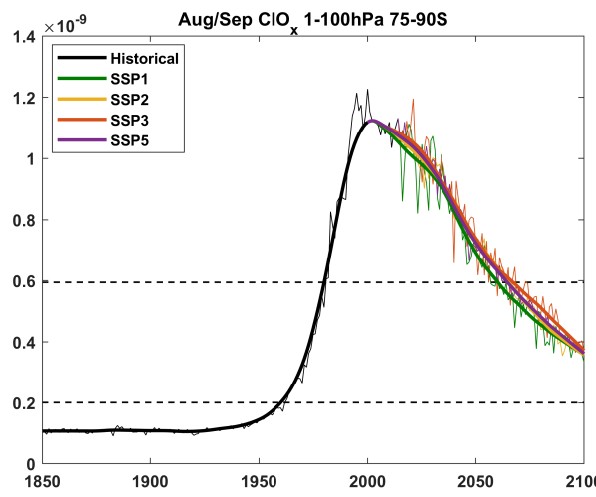

**Figure 2.** Same as Figure 1 for the Aug/Sep mean volume mixing ratio of $ClO_x$ in the Antarctic stratosphere (1-100 hPa).

(Maliniemi et al., 2020). However, after 2050s no major SPEs occur in the CMIP6 solar reference scenario (Maliniemi et al., 2020). Should there be a series of major SPEs in the period between 2050 and 2100 then we would expect the levels of ozone in the Antarctic to be lower than in these projections, further decreasing the likelihood of a full ozone recovery in the Antarctic
upper stratosphere.

Figure 4 shows the relative change of ozone and $ClO_x$ from 1960 to 2000 over the whole atmosphere during Aug/Sep and Feb/Mar. The $ClO_x$ increase is substantial over the whole atmosphere and up to 6 times higher in polar regions in 2000 relative to 1960. During Austral winter strong ozone depletion occurs in the Antarctic lower stratosphere. In addition, there is a 15-25% decrease of ozone in the polar upper stratosphere. This is approximately the altitude of peak effectiveness of Cl/ClO catalytic
cycle that occurs throughout the year in the presence of sunlight, while ClO/ClO cycle has peak effectiveness at the lower stratosphere and requires colder temperatures and the presence of polar stratospheric clouds (Lary, 1997).

### 3.2   Global ozone difference between SSP5 and SSP2 at the end of the 21st century

Figure 5 presents the difference in monthly ozone between SSP5 and SSP2 during 2090-2100. There is substantially more stratospheric ozone in SSP5 relative to SSP2. SSP5 has approximately 0.5 to 1.0 ppm more equatorial stratospheric ozone
above 20 hPa in all months. However, below 20 hPa SSP5 has significantly less equatorial ozone than in SSP2 (up to -0.3 ppm). An additional feature is seen in the mesosphere where consistently lower ozone levels are predicted in SSP5 than in SSP2. However, the negative anomalies are less than -0.1 ppm in regions other than high latitudes.

These global differences between SSP5 and SSP2 can be explained in terms of carbon dioxide and methane emissions (Kirner et al., 2015). The increased ozone in the upper stratosphere is caused by increased ozone production due to a cooler
future middle atmosphere. This is because of the temperature dependency of the Chapman cycle (Brasseur and Solomon, 2005).

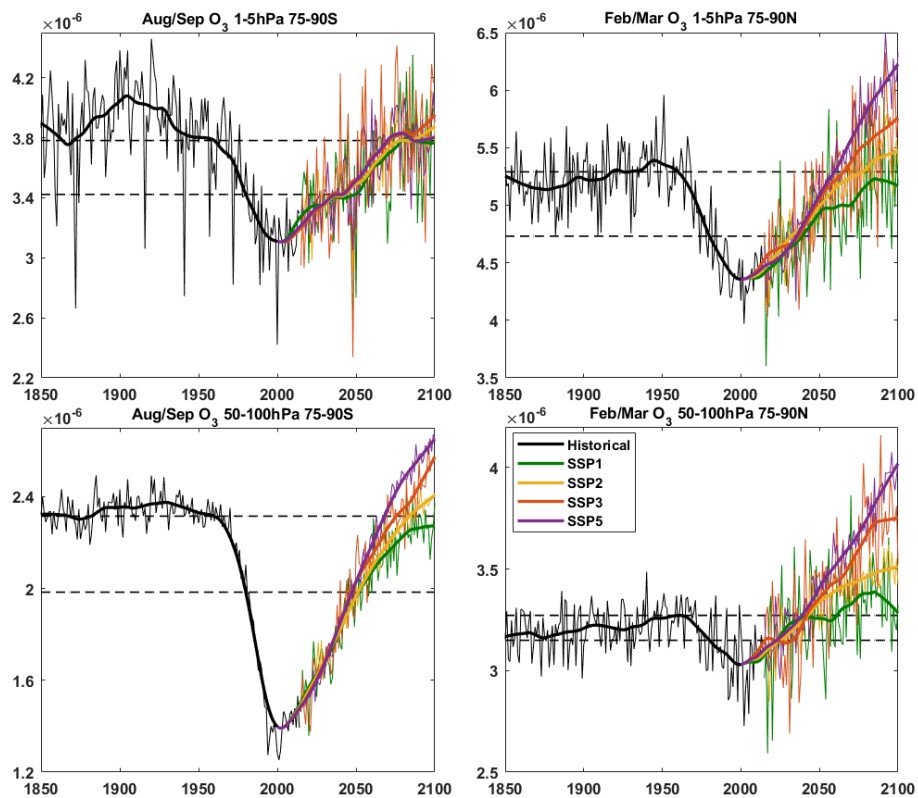

**Figure 3.** Same as Figure 1 for the mean volume mixing ratio of ozone in the Antarctic upper (1-5 hPa; top left) and lower stratosphere (50-100 hPa; bottom left), and in the Arctic upper (1-5 hPa; top right) and lower stratosphere (50-100 hPa; bottom right).

Figure 6 shows the temperature difference between SSP5 and SSP2. The temperature is between 4 to 8 K lower in SSP5 than in SSP2 in the upper stratosphere, with the largest differences in high latitudes during winter. The mesospheric ozone decrease between SSP5 and SSP2 could be partly due to additional methane emissions in SSP5 (Riahi et al., 2017). Methane oxidation produces water vapour and hydrogen oxides ($HO_x$) (le Texier et al., 1988), which Kirner et al. (2015) proposes to influence the

evolution of ozone in the mesosphere.

      Negative ozone anomalies in the equatorial lower stratosphere are mainly due to dynamical changes. Climate change has been predicted to accelerate the Brewer-Dobson circulation (Garcia and Randel, 2008; Butchart, 2014) as shown for the annual BDC in different scenarios in Figure 7. The largest equatorial vertical residual circulation speed at the end of the 21st century occurs in SSP5, followed by SSP3, SSP2 and SSP1, respectively. One can also see that the meridional transport at 50 hPa

altitude accelerates in both hemispheres in the future, and more in SSP5 than in SSP2. This leads to enhanced transport of ozone from the lower equatorial stratosphere, resulting in a negative anomaly in SSP5 relative to SSP2 (Langematz, 2018). The ozone difference in lower equatorial stratosphere between SSP5 and SSP2 could also be partly due to increased overhead ozone, which attenuates the ultraviolet radiation and decreases the photolysis of oxygen in this region (Kirner et al., 2015).

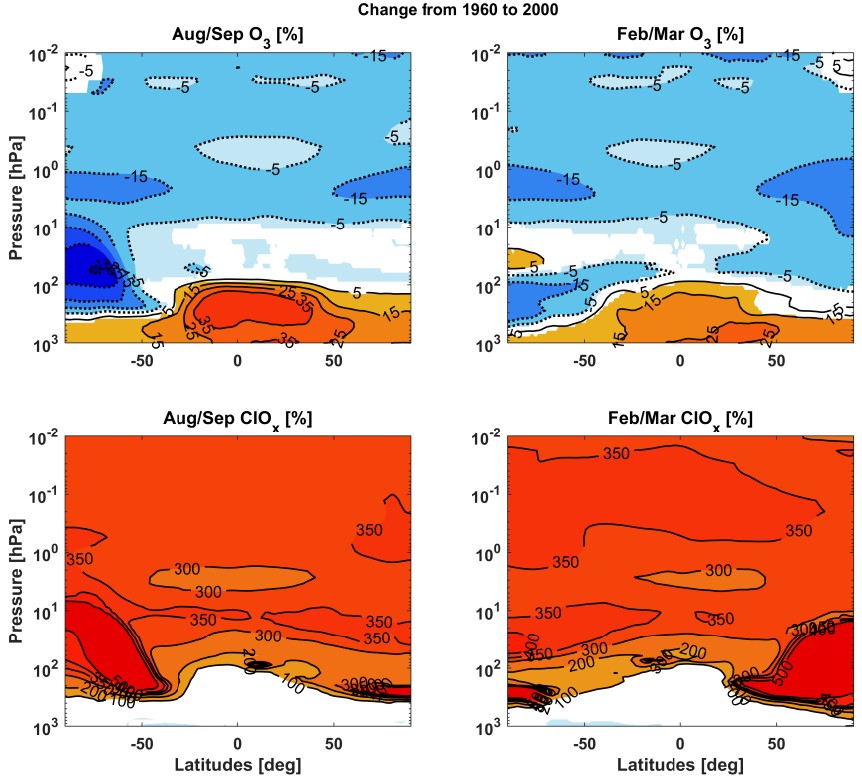

**Figure 4.** Relative change of ozone (Aug/Sep: top left, Feb/Mar: top right) and ClO$_x$ (Aug/Sep: bottom left, Feb/Mar: bottom right) from 1960 to 2000. Positive contour levels for ozone are 5%, 15%, 25% and 35% (solid lines) and negative contour levels -5%, -15%, -25%, -35% and -50% (dotted lines). Positive contour levels for ClO$_x$ are 100%, 200%, 300%, 350%, 400% and 500% (solid lines). Colour shading indicates areas significant at the 95% level calculated with a Mann-Kendall test and a false detection rate.

### 3.3 Polar ozone and NO$_x$ differences between SSP5 and SSP2 at the end of the 21st century

Figure 5 shows that the Arctic stratosphere ozone in SSP5 exceeds ozone in SSP2, reaching the highest values during winter (November to March) but this does not occur in the Antarctic stratosphere. During winter (June to October) a negative ozone anomaly (in SSP5 relative to SSP2) is obtained descending from 1 hPa to 10-20 hPa.

Figure 8 shows the NO$_x$ difference between SSP5 and SSP2 averaged between 2090 and 2100. Over the whole atmosphere there is slightly less NOx in SSP5 than in SSP2. This is in line with slightly lower N$_2$O emissions in SSP5 than in SSP2 (Riahi
et al., 2017) and that a cooler stratosphere increases the chemical destruction of NO$_x$ (Stolarski et al., 2015). However, one can see that there is a substantial increase of NO$_x$ in the Antarctic mesosphere and upper stratosphere from June until September. The NO$_x$ increase in the upper stratosphere is up to 10 parts per billion (ppb). Maliniemi et al. (2020) showed that southern polar mesospheric descent rates will accelerate in the future under higher greenhouse gas forcing, which leads to more NO$_x$

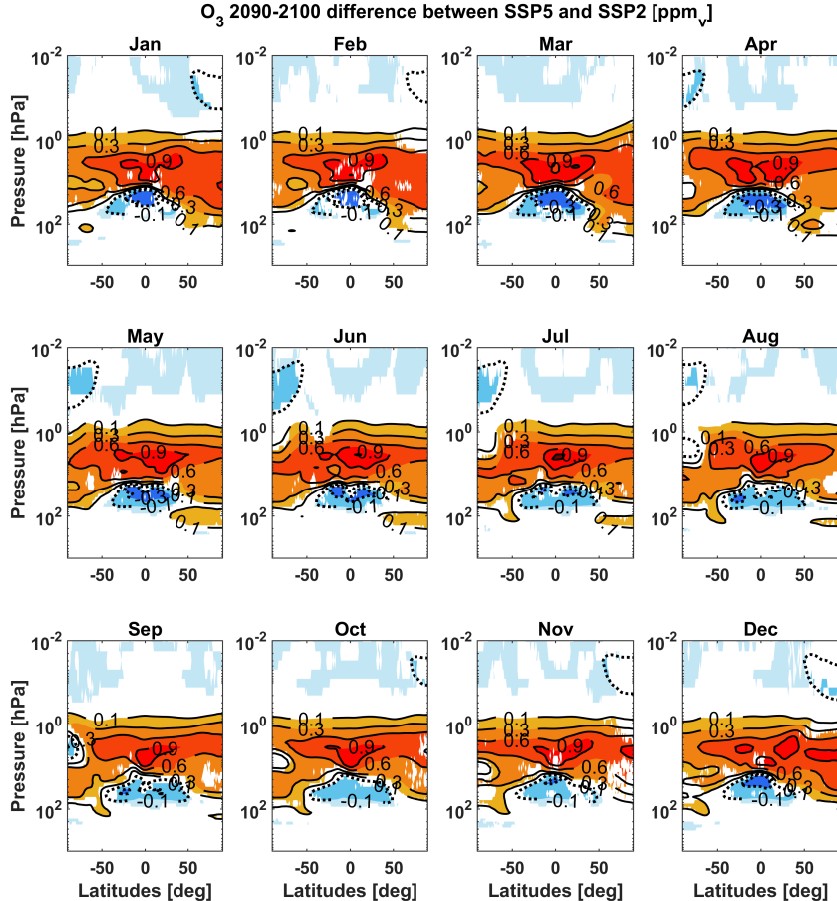

**Figure 5.** The difference in the monthly zonal mean ozone between SSP5 and SSP2 in parts per million during 2090-2100. Positive contour levels are 0.1, 0.3, 0.6, 0.9 and 1.2 ppm (solid lines), and negative contour levels are -0.1, -0.3 ppm (dotted lines). Colour shading indicates areas significant at the 95% level calculated with a monte carlo simulation and a false detection rate.

being transported from the upper mesosphere/thermosphere to the upper stratosphere. In SSP5 there is about a 10-20% faster
descent at the end of the 21st century than in SSP2 (Maliniemi et al., 2020).

The $NO_x$ difference in the northern hemisphere is less dramatic. There is an increase in the upper mesosphere from November to March (see Figure 8) but it does not descend to lower altitudes. Figure 6 shows the difference of zonal wind between SSP5 and SSP2 during southern and northern winters. The polar vortex is weaker in the northern hemisphere but slightly stronger in the southern hemisphere in SSP5. A stronger polar vortex tends to accelerate mesospheric descent due to the fil-
tering of westerly gravity waves and the resulting easterly gravity wave drag in the mesosphere. As a result, $NO_x$ anomalies descend further downward in the southern hemisphere.

Figure 9 shows polar ozone and $NO_x$ differences between SSP5 and SSP2 during the winter months in both hemispheres. The altitude of the negative ozone anomaly in the Antarctic stratosphere follows the altitude of $NO_x$ increase closely and is

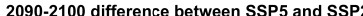

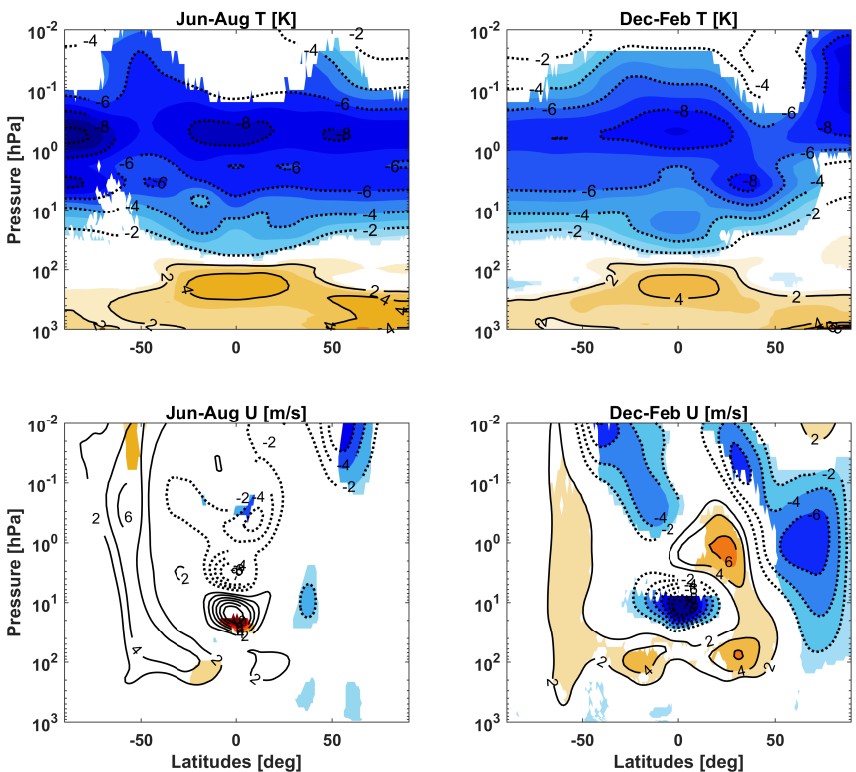

**Figure 6.** Difference in the zonal mean temperature (top left: Jun-Aug, top right: Dec-Feb) and zonal mean zonal wind (bottom left: Jun-Aug, bottom right: Dec-Feb) between SSP5 and SSP2 during 2090-2100. Positive contour levels are 2, 4, 6, 8 and 10 (solid lines), and negative contour levels are -2, -4, -6, -8 and -10 (dotted lines) as K for temperature and m/s for zonal wind. Colour shading indicates areas significant at the 95% level calculated with a monte carlo simulation and a false detection rate.

statistically significant during September. In the northern hemisphere winter no polar $NO_x$ increase occurs below 0.1 hPa, and ozone concentration in the stratosphere does not experience any dramatic variability over different winter months. One can also see that after the $NO_x$ peak has passed in the Antarctic, the ozone values around 1 hPa during October return back to higher levels in SSP5 than in SSP2 and become comparable to the ozone levels in the Arctic stratosphere at the same altitude.

Transport to the polar region at 1 hPa is primarily from above during winter (Smith et al., 2011), while in the lower polar stratosphere meridional transport from the equatorial lower stratosphere via the BDC is important. Ozone super recovery in the upper polar stratosphere is thus mainly predicted due to the increased production (WMO, 2018, Chapter 4), while in the lower polar stratosphere it is because of the increased transport from the equatorial lower stratosphere (Langematz, 2018). While our simulation study is not a single forcing experiment and thus not optimal to precisely estimate different contributions, they do present self-consitent projections of the future evolution of ozone. Enhanced transport of $NO_x$ to the Antarctic upper

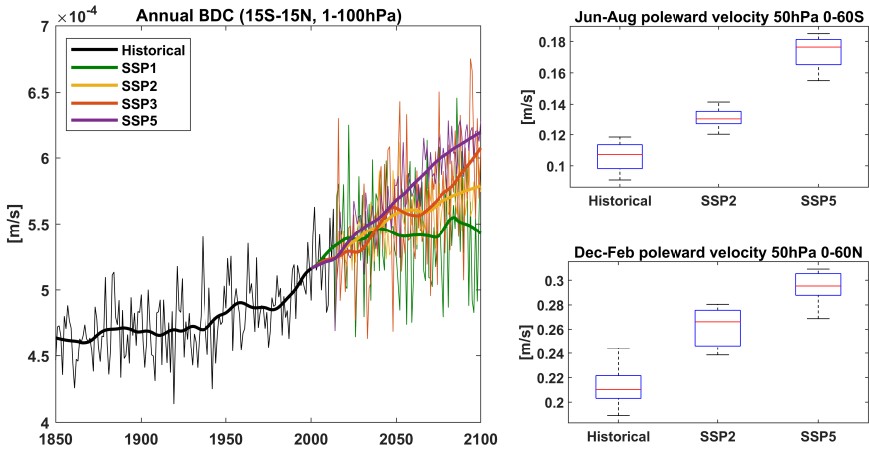

**Figure 7.** Left: The time series of the annual Brewer-Dobson circulation (vertical residual circulation speed at $15°S$-$15°N$, 1-100hPa). Colours represent the same simulations as in Figure 1. Right: Boxplot of southern (top) and northern (bottom) hemisphere meridional residual circulation speed during local winter at 50hPa. Data for historical period in the boxplot is for an average from 2004 to 2014 and for SSP periods between 2090 and 2100. Red lines represent median, blue boxes represent 25-75 percentiles and grey dotted lines the total range.

stratosphere from above as a result of climate change could counteract enhanced ozone production seen elsewhere in the
atmosphere, and potentially prevent an ozone super recovery in the Antarctic upper stratosphere (see Figure 2).

## 4 Summary

In this paper we show that future scenarios with stronger greenhouse gas forcing lead to overall higher levels of simulated stratospheric ozone. Ozone in SSP5 relative to SSP2 is higher in the low and mid-latitudinal upper stratosphere at the end of the 21st century. This is a consequence of increased greenhouse gas emissions and the resulting lower temperatures in the middle
atmosphere. A cooler stratosphere will increase ozone production, leading to an ozone increase in the upper stratosphere. SSP5 has less ozone than SSP2 in the equatorial lower stratosphere. This negative ozone anomaly is a consequence of accelerated transport to the polar lower stratosphere via a stronger Brewer-Dobson circulation.

In SSP3 and SSP5, ozone will have a super recovery in the Arctic stratosphere and Antarctic lower stratosphere towards 2100, in agreement with WMO (2018, Chapter 4). However, ozone in the Antarctic upper stratosphere reaches similar levels
across the different future scenarios which are not above the pre-CFC levels at the end of the 21st century. We show that this is due to excess $NO_x$ descending to the upper stratosphere from the polar thermosphere and upper mesosphere in the stronger greenhouse gas scenarios (Maliniemi et al., 2020) and the resulting catalytic ozone loss.

Following the adoption of the Montreal protocol, stratospheric $ClO_x$ will decrease in the future (Velders et al., 2007). As a result, the catalytic $NO_x$ cycle is more important for ozone variability in the future. Polar thermospheric and upper meso-

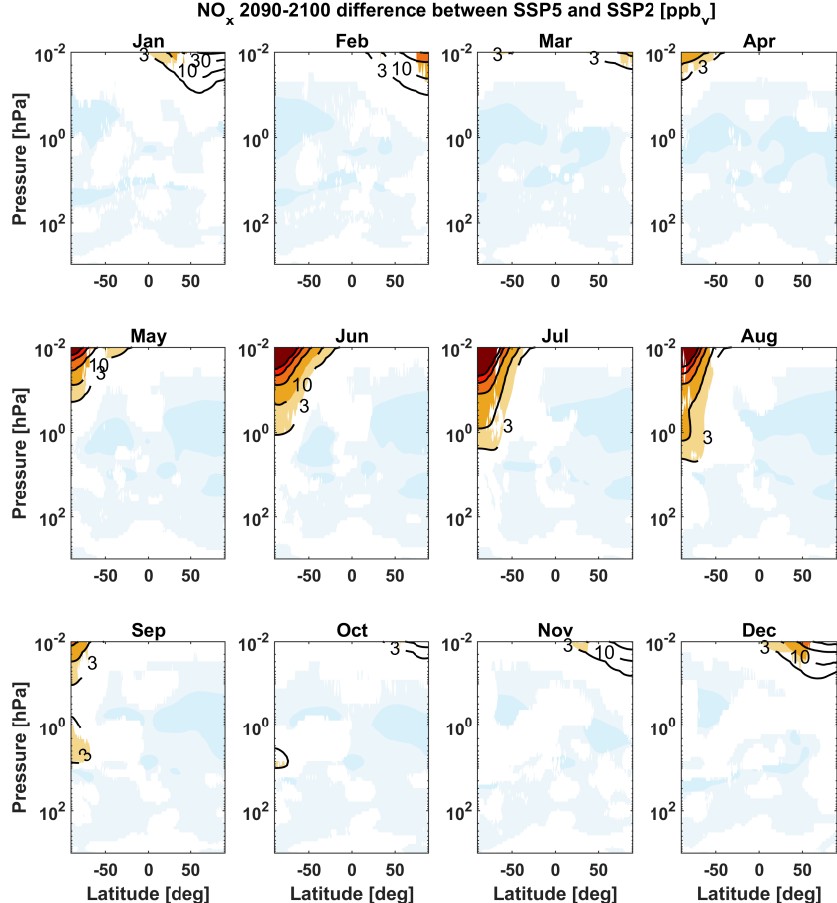

**Figure 8.** Monthly zonal mean NO$_x$ difference between SSP5 and SSP2 in parts per billion during 2090-2100. Positive contour levels are 3, 10, 30, 60 and 90 ppb (solid lines). Colours represent areas significant at the 95% level calculated with a monte carlo simulation and a false detection rate. All negative responses are below -1 ppb.

spheric NO$_x$ is mainly produced by EEP and partly by solar UV via transport from low latitudes (Gérard et al., 1984). During winter polar darkness, NO$_x$ has a long chemical lifetime and descends to the stratospheric altitudes. Since the descent rate is accelerating in the Antarctic mesosphere under higher greenhouse gas emissions, this indirect NO$_x$ will have an increasing importance for the future of ozone in the Antarctic stratosphere.

Seasonal stratospheric ozone depletion due to the descending indirect NO$_x$ has been also shown to influence stratospheric temperatures and the polar vortex (Arsenovic et al., 2016; Salminen et al., 2019; Asikainen et al., 2020). The polar vortex is connected to the tropospheric annular mode and has a notable effect on weather in the mid- and high latitudes (Thompson and Solomon, 2002; Kidston et al., 2015). Thus, there is a great potential of improving future projections (not only to ozone) and

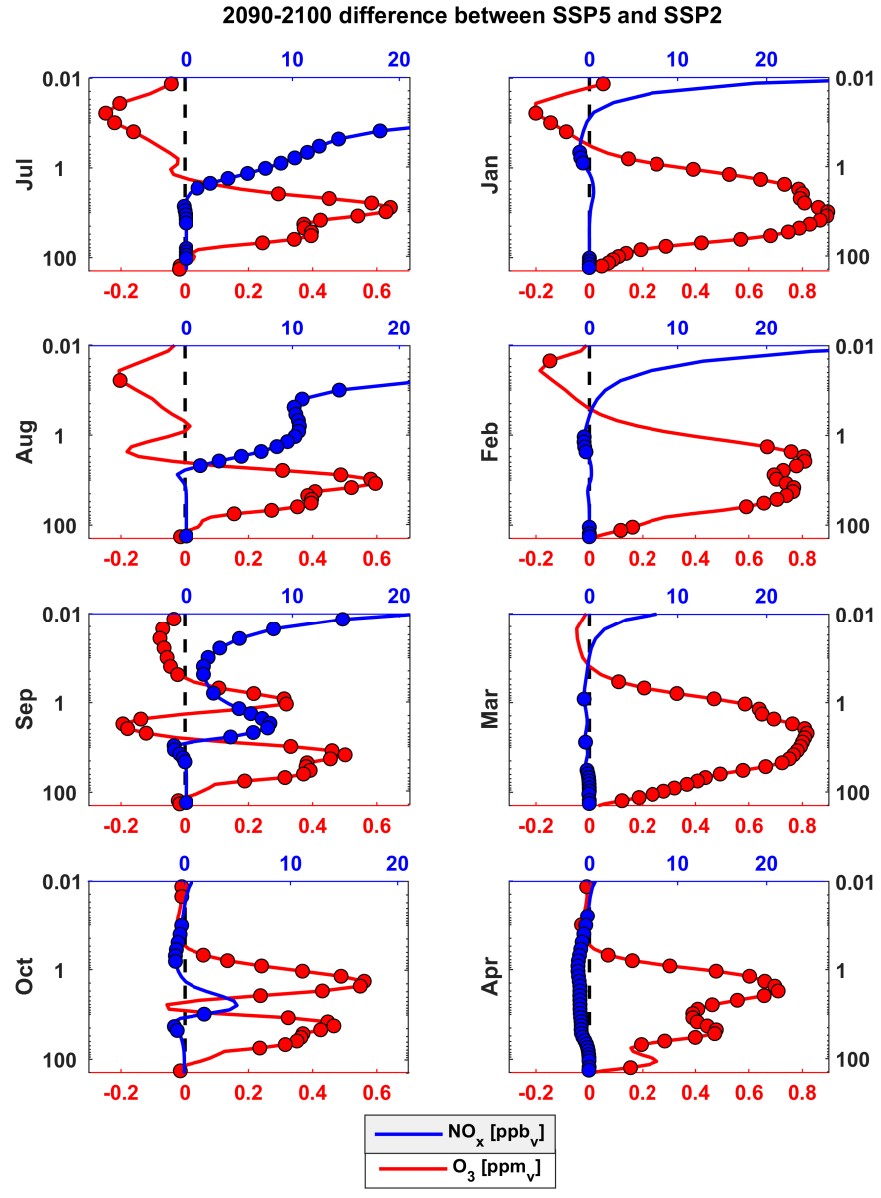

**Figure 9.** Monthly zonal mean polar (75-90°S during Jul-Oct and 75-90°N during Jan-Apr) $NO_x$ (blue, parts per billion) and ozone (red, parts per million) differences between SSP5 and SSP2 during 2090-2100. Vertical axes show the altitude in air-pressure units from 0.01 to 200 hPa. Coloured circles represent values significant with 95% calculated with a monte carlo simulation and a false detection rate.

seasonal forecasts of polar regions by implementing a more accurate solar forcing, including EEP to the earth system models (Matthes et al., 2017).

*Data availability.* WACCM simulations used in this study are available as part of the CMIP6 on the Earth System Grid (https://esgf-node.llnl.gov/projects/cmip6/).

*Author contributions.* D.R.M provided the WACCM model outputs. V.M analysed the data and wrote the manuscript. All authors contributed to the analyses of the results and modification of the manuscript.

*Competing interests.* The authors declare no competing interest.

*Acknowledgements.* We thank the National Center for Atmospheric Research for WACCM model outputs. The research has been funded by the Norwegian Research Council under Contracts 223252/F50 (BCSS) and 300724 (EPIC). D.R.M. was supported by the NSF (#1650918). The National Center for Atmospheric Research is sponsored by the National Science Foundation.

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
