# Peer review of "Effects of enhanced downwelling of $NO_x$ on Antarctic upper stratospheric ozone in the 21st century"

_Atmospheric Chemistry and Physics, 2021_

## Author Comment (AC1)

**Reply to reviewers:**

We would like to thank both reviewers for careful reading of paper and valuable comments. For convenience, the original comments by reviewers are indicated below in **bold blue font**. Our response to each comment is given in normal font.

Comments for anonymous referee (RC1):

**Maliniemi et al. show in their manuscript WACCM model simulations for different greenhouse gas scenarios over the 1850 – 2100 time frame with a focus on NOx descent from the MLT region into the stratosphere and its effect on polar stratospheric ozone. Their main conclusion is, that due to the enhanced descent of NOx in the SH high latitudes, ozone super recovery will not take place over the Antarctic. The study is well conceived, the results clear and convincing. This is a nice focussed study that should be published in Atmos. Chem. Phys. The paper is well written and I have only a few minor specific comments (below) that should be taken into account before publication. I am, however, unsettled regarding the title: "Ozone super recovery cancelled in the Antarctic upper stratosphere", which is a bit of a catchy phrase. I guess the meaning is the counter-acting effect of NOx enhancements that cancels super-recovery? Maybe better say so. As Johnson says "where ever you meet with a passage which you think is particularly fine, strike it out" …**

We agree that the title was a bit too general and catchy to explain the work that is presented. We changed the title as follows: Effects of enhanced downwelling of NOx on Antarctic upper stratospheric ozone in the 21$^{st}$ century.

**Specific comments**

**L23: "especially from equatorial lower stratosphere" sounds strange, as ozone is not predominantly transported directly from the equatorial lower stratosphere to high latitudes. Suggestion: "…leads to enhanced transport of ozone to high latitudes, and a reduction of ozone in the equatorial lower stratosphere…" (btw "enhanced" was also spelled wrong)**

The suggested phrase is now used.

**L35: maybe you can spend a few words, why the descent will be stronger in a stronger vortex. From a dynamical point of view, the opposite may be expected, I believe? Are you referring to a stronger apparent descent of tracers, because of reduced meridional mixing, or is also w-bar-star increasing?**

Vertical residual circulation ($\overline{w^*}$) is indeed strengthening in the southern polar mesosphere. Our earlier paper (Maliniemi et al., GRL, 2020) showed faster descent in scenarios with higher greenhouse gas emissions (Figure 2 in Maliniemi et al., 2020, also shown below). It is correct that stratospheric residual circulation, which is driven by planetary wave drag, is weaker during stronger polar vortex. This is because stronger westerly wind reflects planetary waves away from the high latitudes. However, mesospheric residual circulation is driven by gravity waves. Easterly directed gravity waves are less filtered by westerly stratospheric zonal wind than westerly directed gravity. When the polar vortex is strong, easterly directed gravity wave drag dominates in the mesosphere driving poleward and downward residual circulation.

[Figure]

**Jun-Sep W* 0.01-1 hPa 70-90S**

**L49: "the mean of ensemble members" = "ensemble mean", or does this mean something different?**

We now use "ensemble mean".

**L63: My understanding of LOWESS (or LOESS) is that this is a regression method. The abstract of Cleveland & Devlin (1988) states: "loess, is a way of estimating a regression surface through a multivariate smoothing procedure, fitting a function of the independent variables locally and in a moving fashion analogous to how a moving average is computed for a time series." However, as I understand, here you have just used a moving average on the time series? Please provide more details on the method you applied.**

We use LOWESS method when calculating the 31-year smoothed timeseries. In this method a smoothed value $x_i$ (corresponding to time $t_i$) of the time series is calculated by a weighted linear least squares regression of the respective data point and its neighboring points $x_{i+k}$, where $k = -15, \ldots, 15$. The length of the smoothing window is constant for all data so that, e.g., in the end points of the time series the regression includes points corresponding to where $k = 0, \ldots, 30$ (series start) or where $k = -30, \ldots, 0$ (series end). The weights are given by the tri-cube weighting function $w_k = \left(1 - \left|\frac{t_i - t_{i+k}}{d(t_i)}\right|^3\right)^3$, where $d(t_i) = \max(t_i - t_{i+k})$ is the maximum time difference between $x_i$ and data points within the smoothing window. The LOWESS method behaves more smoothly than a regular running mean, especially in the end points of the data series. See figure below for an example. We also included additional reference to one earlier paper which gives same details as given above (Maliniemi et al., 2014).

[Figure]

**L150: "It is clear that stratospheric ClO$_x$ will decrease in the future, following the adoption of the Montreal protocol": It is not precisely clear to me what the meaning of this sentence is. Do you just mean "Following the adoption of the Montreal protocol, stratospheric ClO$_x$ will decrease in the future"? Or: "Stratospheric ClO$_x$ will decrease in the future, if the Montreal protocol is adhered to"?**

The sentence now reads as "Following the adoption of the Montreal protocol, stratospheric ClOx will decrease in the future".

**L152: "following winter darkness when its chemical lifetime is long": why "following"? The lifetime is longest "during winter darkness", not "following winter darkness", or do I misunderstand something here?**

This was wrong wording. We replaced "following" with "during".

**L157: "effect on winter weather" is not exactly true: Previous studies showed the largest effect of the Antarctic vortex on SH surface during December, which is mid-summer.**

We rewrote the sentence as: "The polar vortex is connected to the tropospheric annular mode which has a notable effect on weather in the mid- and high latitudes."

**Technical corrections**

**L91: "there are" -> "there is"**

We corrected this.

**L137: greenhouse**

It is now corrected.

**References:**
Maliniemi, V., Asikainen, T. and Mursula, K. (2014), Spatial distribution of Northern Hemisphere winter temperatures during different phases of the solar cycle, J. Geophys. Res. Atmos., 119, doi: 10.1002/2013JD021343

Maliniemi, V., Marsh, D.R., Nesse Tyssøy, H. and Smith-Johnsen, C. (2020), Will climate change impact polar NOx produced by energetic particle precipitation?, Geophys. Res. Lett., 47, doi:10.1029/2020GL087041

Comments for Susan Solomon (CC1/RC2):

**This paper uses WACCM CMIP6 simulations of the 21st century to examine the future behavior of ozone in the upper stratosphere using different scenarios. The primary conclusion is that while a cooler stratosphere leads to ozone increases over much of the upper stratosphere, an exception occurs in the Antarctic, where the authors argue that increased downwelling within the strong polar vortex deposits greater abundances of NOx from EEP, which deplete ozone. The paper makes some interesting points but requires revisions before publication. My comment are as follows:**

**1. The paper doesn't show what happens to the total ozone column, which is a key quantity. Because much of the ozone loss is in the lowermost stratosphere (especially in the Antarctic), I would suspect that Antarctic total ozone still undergoes recovery, perhaps even super-recovery. Please add a figure showing what happens to the global total ozone as well in this model.**

We now show total column ozone (TCO) for southern and northern polar regions (75-90° S/N) (new Figure 1). The figure is also shown below. As you mentioned most of the ozone loss occurs in the lower stratosphere and TCO follows the lower stratosphere ozone volume mixing ratio closely.

[Figure]

**2. The use of the term 'recovery' or 'super-recovery' in this paper raises the question of recovery from what – it should be recovery from ozone depletion due to CFCs if this term is to be used. Figure 1 shows values of average ozone mixing ratios from 1-10 hPa and from 10-100 hPa in different months and regions. These are very broad swaths, and the ozone depletion will occur only where chlorine is important; it may be small or even near zero at some of these levels. Also, the mixing ratio at 10 hPa is much larger than at 1 hPa, so this quantity is heavily weighted towards lower altitudes and is hard to interpret. The problem is even worse in Figure 2, where ClOx abundances for 1-100 hPa are presented. I suspect that there was not much ozone depletion from 1960-2000 (if any) at for example 1 hPa, so it's not clear that super-recovery is the right word there…it may not have been substantially "depleted" in the first place. Please provide an additional figure showing contour plots of the changes in ozone and ClOx from -90 to 90 degrees, 1000 to 0.01 hPa, for 1960-2000 and clarify where depletion occurs versus where the ozone changes of interest here occur. If there are regions or months where little or no ozone depletion occurred from 1960-2000 in the first place, then recovery or super-recovery is the wrong language and should be altered. These plots should probably be in percent units rather than mixing ratio, so that we can see the extent of changes everywhere.**

We slightly modified the altitude limits for Figure 1 (new Figure 3). The upper stratosphere is now 1-5 hPa and the lower stratosphere is 50-100 hPa. In addition, we added a figure (new Figure 4) showing Aug/Sep and Feb/Mar ozone and ClOx change from 1960 to 2000 over the whole atmosphere. Significance estimates were done with the Mann-Kendall trend test and multiple hypothesis test. The

new figures are also shown below. As one can see there is substantial ozone depletion between 1 and 5 hPa (15-25% decrease). ClOx increases are several hundred precent everywhere above 100 hPa. The southern polar region has up to 500% more ClOx below 10 hPa and 350% to 400% more ClOx between 1 and5 hPa. Ozone depletion between 1 and 5 hPa coincides with the peak altitude of Cl/ClO catalytic cycle effectiveness in Lary (1997), whereas at lower altitudes the dominant catalytic chain is likely ClO/ClO which requires colder temperatures and polar stratospheric clouds.

[Figure]

**2. Please elaborate what is in the model with regard to solar proton events, and whether those could modify the picture. See Stone et al., GRL, 2018.**

The CMIP6 solar reference scenario has scenario for SPEs. One can see Jul-Sep average timeseries of SPEs during 1850-2100 in Figure 2 in Maliniemi et al. (2020) (also shown below). SPE influence can be seen in Antarctic upper stratospheric ozone evolution during 1850-2100 (see figure in previous comment) as large spikes in yearly values (thin lines), e.g., during 2000 or around 2050. However, SPE scenario has no large SPEs after 2060, and thus SPEs have negligible influence on our analysis of difference between SSP5 and SSP2 during 2090-2100.

[Figure]

**3. Figure 6 shows that the maximum NOx increase occurs in August. It is essentially gone by October. Please explain why this occurs. I would have expected a longer residence time. Is it being mixed out of the vortex? Chemically destroyed? Or?**

Chemical destruction of NOx in sunlit will be faster in the future with cooler stratosphere (Stolarski et al., 2015). During October mesospheric NOx will be mostly destroyed by photolysis before it reaches stratospheric altitudes. Thus, residence time after polar night ends will be shorter in SSP5 compared to SSP2.

**4. There are many mistakes in basic English in the paper. I note that at least one of the authors is a native speaker of the English language and request that attention be paid to proper English to make the paper more readable. "The" is missing in many places, for example, and those mistakes should be fixed by the authors, not the reviewers.**

The grammar has been improved in the revised manuscript.

**5. The title seems unclear, for the reasons noted above. Please rephrase. Something like "Effects of enhanced downwelling of NOx on Antarctic upper stratospheric ozone in the 21st century" might be suitable.**

We agree that the title was a bit too general and catchy to explain the work that is presented. We changed the title as follows: Effects of enhanced downwelling of NOx on Antarctic upper stratospheric ozone in the 21st century.

**6. Line 22 and later (e.g., 106). Garcia and Randel 2008 was a modelling study; Butchart et al. 2014 was a review but stated what while models showed BDC increases, the data was unclear. The statement that the strength of the BDC is increasing should be edited if it is just based on models; if it's based on data then please provide an appropriate reference to back it up.**

Yes, it is based on models. The text now reads: "Climate models predict that the Brewer-Dobson circulation (BDC) is increasing (Garcia and Randel, 2008; Butchart, 2014) and leads to enhanced transport of ozone to polar lower stratosphere, and a reduction of ozone in the equatorial lower stratosphere." and "Climate change has been predicted to accelerate the Brewer-Dobson circulation…"

**7. Line 27. Where in WMO, 2018, and does this pertain to ozone in the upper stratosphere?**

We now mention that it is in Chapter 4 (Polar Stratospheric Ozone) of the WMO (2018) report. In addition, we noticed in the report that it is mainly transport to the polar lower (not upper) stratosphere that is causing dynamical changes for future ozone. It is shown in the report (Section 4.5.3.3 and Figure 4.20) that late winter upper stratospheric ozone projections are roughly similar in the Arctic and in the Antarctic, while lower stratosphere projections differ substantially. Thus, we have modified the text as: "This transport effect is projected to be stronger in the northern hemisphere, leading to a more prominent ozone super recovery in the Arctic lower stratosphere than in the Antarctic lower stratosphere."

**8. Line 28-29. How can you be sure that EEP are the main cause? Please note the existence of other sources of NOx (SPE, but also simply downward transport from other sources besides EEP). Please clarify how it is you know that the NO increase you calculate is indeed due to EEP (versus e.g., photolysis and photoionization reactions at higher altitudes, transport from higher altitudes but originating from lower, sub-auroral latitudes, etc.); also are you referring to auroral electrons or others? If you can't be sure that EEP are the cause of the NOx change, please use different language. You could say uppper mesospheric/thermospheric NOx for example, if transport from higher altitudes and lower latitudes is important.**

It is correct that without a single forcing experiment it can only be suggested that it is due to EEP. Gerard et al. (1984) shows that there is roughly an order of magnitude more polar thermospheric odd nitrogen in simulation with EEP, solar UV and transport compared to simulation with just solar UV and transport. Our earlier paper (Maliniemi et al., 2020) also showed that the EEP is a dominating source during historical period 1850-2014 in the mesosphere (Supplementary material in Maliniemi et al., 2020). However, we cannot neglect the possibility that transport from low latitudes in the future becomes more important. This is an interesting study question which will be further analyzed. However, we feel that this is out of the scope of this paper. Consequently, in the manuscript we now talk about upper mesospheric/thermospheric NOx more generally.

**9. Line 55. Briefly describe what was done in the CMIP6 recommendations on solar activity that are relevant here.**

We use CMIP6 solar reference scenario, i.e., a realistic scenario of solar activity level during the 21$^{st}$ century. The forcing includes total solar and solar spectral irradiance as well as magnetospheric electrons, solar proton events and galactic cosmic rays. Scenario is constructed based on the estimate that solar activity will return to early 20$^{th}$ century levels during the 21$^{st}$ century (Matthes et al., 2017). We have modified the description in the manuscript.

**10. Line 63. Why 31 years?**

We wanted the LOWESS-method to have at least 30-year smoothing window (for a typical climate mean) and an equal number of data points before and after the specific data point in question. Thus, we have 31-year smoothing window (x-15…x…x+15).

**11. Figure 1. I don't understand why these figures are so spikey if they are treated with the 31 year mean/LOWESS approach as described. Please explain.**

In the figure caption it is mentioned that: Thin lines represent yearly average and thick lines represent 31-year smoothed trend calculated with LOWESS-method. Thus, the spikes in thin lines occur because there is no long-term smoothing. Thick lines represent 31-year LOWESS smoothing and evolve without spikes.

**12. Figure 2. Can you explain why the variability is greater in the future than the past?**

The historical simulation has 3 ensemble members, SSP1 and SSP3 have 1 ensemble member, and SSP2 and SSP5 have 5 ensemble members. Shown results are ensemble means for historical, SSP2 and SSP5 model runs (also mentioned in the manuscript). One can see that in Figure 2 the variability in thin lines (yearly averages) is larger only in green and red lines, i.e., in SSP1 and SSP3. We also mention this issue now in the manuscript.

**13. Line 97. The increased ozone throughout the upper stratosphere is likely caused mainly by colder temperatures, not just at the equator.**

That is true. We now say: The increased ozone in the upper stratosphere…

**14. Lines 103-104. I don't think this statement is quite right. While it seems likely that HOx production is important in the mesosphere, a key point is that mesospheric ozone loss chemistry is much less temperature sensitive than that at the stratospause – different reactions with different energies of activation are involved. Please rephrase.**

We now write it as: "Mesospheric ozone decrease between SSP5 and SSP2 could partly be due to additional methane emissions in SSP5 (Riahi et al., 2017). Methane oxidation produces water vapour and hydrogen oxides (HOx) (le Texier et al., 1988), which Kirner et al. (2015) proposes to influence the evolution of ozone in the mesosphere.

**15. Lines 130-135. Here the authors raise some doubt about how much of the changes referred to are indeed due to EEP and NOx versus other causes (e.g., dynamical). This is quite concerning since it's the main conclusion of the paper! Can dynamical contributions be checked by looking at other quantities in the model? for example, you might use SF6 as a tracer for downward mesospheric transport and see how percent changes in SF6 compare to the percent changes in ozone, for example. A clearer analysis is needed to support the paper's conclusions, or the points made earlier and in the conclusions need changes.**

We have revised this part in the manuscript. As explained in the question 7, the manuscript was misleading regarding the effect of transport to upper stratosphere. Most of the transport to the polar stratosphere around 1 hPa altitude during winter is downward motion from the mesosphere (Smith et al., 2011), while in the lower stratosphere meridional transport from the equatorial stratosphere to

polar region occurs (Kirner et al., 2014). Figure below shows the vertical residual circulation ($\overline{w^*}$) for the polar mesosphere in different SSPs (Maliniemi et al., 2020). One can see that at the end of the 21st century, SSP5 (purple) has about 10-20% faster descent rate than SSP2 (yellow). Figure 7 (new Figure 9) in the manuscript also shows clearly that the negative ozone anomaly occurs exactly at the altitude of NOx increase, while in Oct (after the NOx peak is below 1hPa) ozone again increases substantially at 1 hPa and returns to the level comparable to the Arctic. We also modified Figure 5 (new Figure 7) and added boxplots for meridional residual circulation ($\overline{v^*}$) speed at 50 hPa in both hemispheres during local winter (historical during 2004-2014, SSPs during 2090-2100). This shows that the lower stratosphere meridional transport in both hemispheres accelerates more in SSP5 than in SSP2. However, we note that it is not optimal to estimate the exact dynamical contributions without a single forcing experiment which is now discussed in the manuscript.

[Figure]

[Figure]

**16. Presumably if the NOx changes are caused by increased downwelling, then they should smoothly increase with time through the time of the simulation. Please show that. Also show what is happening at higher levels (e.g., 100-120 km) and lower latitudes to address my earlier comment on the potential importance of poleward and downward transport.**

Upper stratospheric NOx time series has been shown in our previous paper (Maliniemi et al., 2020). A figure from that paper is also shown below. As mentioned in question 8, we now discuss upper mesospheric/thermospheric NOx more generally and note the existence of transport from low latitudes related to solar UV. Concerning higher altitudes (100-120km), we feel that this is out of the

scope of this paper which concentrates on stratospheric ozone in the future. Thermospheric changes between different SSPs will be analyzed in the future work.

[Figure]

17. **Line 140. Earlier, you said the equatorial lower stratospheric changes were dynamical.**

Yes, this part was a bit confusing. The manuscript now reads: "Ozone in SSP5 relative to SSP2 is higher in the low and mid-latitudinal upper stratosphere at the end of the 21st century. This is a consequence of increased greenhouse gas emissions and the resulting lower temperatures in the middle atmosphere. A cooler stratosphere will increase ozone production, leading to an ozone increase in the upper stratosphere. SSP5 has less ozone than SSP2 in the equatorial lower stratosphere. This negative ozone anomaly is a consequence of accelerated transport via a stronger Brewer-Dobson circulation."

**References:**

Gérard, J.-C., Roble, R. G., Rusch, D. W., and Stewart, A. I. (1984), The global distribution of thermospheric odd nitrogen for solstice conditions during solar cycle minimum, *J. Geophys. Res.*, 89, 1725– 1738, doi:10.1029/JA089iA03p01725

Kirner, O., Ruhnke, R., and Sinnhuber, B.-M. (2015): Chemistry–climate interactions of stratospheric and mesospheric ozone in EMAC long-term simulations with different boundary conditions for CO2, CH4, N2O, and ODS, Atmosphere-Ocean, 53, 140–152, https://doi.org/10.1080/07055900.2014.980718

Lary D. J. (1997), Catalytic destruction of stratospheric ozone, *J. Geophys. Res.*, 102, 21515– 21526, doi:10.1029/97JD00912

Maliniemi V., Marsh, D.R., Nesse Tyssøy, H., and Smith-Johnsen, C. (2020), Will climate change impact polar NOx produced by energetic particle precipitation?, Geophys. Res. Lett., 47, doi:10.1029/2020GL087041

Matthes, K., Funke, B., Anderson, M., Barnard, L., Beer, J., Charbonneau, P., Clilverd, M., Dudok de Wit, T., Haberreiter, M., Hendry, A., Jackman, C., Kretschmar, M., Kruschke, T., Kunze, M., Langematz, U., Marsh, D., Maycock, A., Misios, S., Rodger, G., Scaife, A., Seppälä, A., Shangguan, M., Sinnhuber, M., Tourpali, K., Usoskin, I., van de Kamp, M., Verronen, P., and Versick, S. (2017): Solar forcing for CMIP6 (v3.1), Geosci. Model Dev., 10, https://doi.org/10.5194/gmd-10-2247-2017

Smith, A. K., Garcia, R. R., Marsh, D. R., and Richter, J. H. (2011), WACCM simulations of the mean circulation and trace species transport in the winter mesosphere, *J. Geophys. Res.*, 116, D20115, doi:10.1029/2011JD016083

Stolarski, R. S., Douglass, A. R., Oman, L. D., and Waugh, D. W.: Impact of future nitrous oxide and carbon dioxide emissions on the stratospheric ozone layer, Environ. Res. Lett., 10, doi:10.1088/1748-9326/10/3/034011, 2015.

---

## Author Response (AR2)

**Reply to reviewer:**

We thank the reviewer for reading of paper and pointing a few additional clarifications. For convenience, the original comments by the reviewer are indicated below in **bold blue font**. Our response to each comment is given in normal font.

**Overall, the authors have done an excellent job responding to my comments and the paper is very much improved. I have only a few remaining concerns, only one of which is major.**

**1) Major. The authors have added some helpful figures on the vertical structure of ozone change, but I think there is still a need to see contour plots of the future ozone changes (latitude-height contour) that are the key point of the paper. I realize there are multiple scenarios, and am not suggesting they all be shown; perhaps . I think what is still needed is a plot for the 21st century change perhaps just SSP5. Figure 4 should be modified to do that. There is little point in showing the percent changes in ClOx in the bottom row. These should be replaced with contour plots for the percentage changes from 2000-2100 like those in the top row, so that we can see the full structure of the effect that is of interest here.**

We modified Figure 4 as recommended. One can see the updated figure below. We also modified the part in the chapter 3.1 where Figure 4 is discussed.

[Figure]

**2) Line 31-34. There were many papers on the topic of NOx transport from the upper atmosphere to the stratosphere via the winter polar vortex and impacting ozone predating the ones given. The original references are, as far as I know, work by Solomon et al. in JGR in 1981 and Brasseur and colleagues around the same time. A very good early review is Garcia, Advances in Space Research, 1992.**

We included citations to Solomon et al. (1981) and Garcia (1992).

**3) Line 91, suggest "which is more notable in the northern hemisphere, as explained further below."**

The suggested phrase is now used.

**4) Line 165. Do you really mean increased production? Or slower loss due to colder temperatures affecting the rate limiting steps in key catalytic cycles.**

Yes, this was misleading. We now write it as: " Ozone super recovery in the upper polar stratosphere is thus mainly predicted due to the decreased ozone loss reactions in colder temperatures..." Similar changes are also made in other parts of the manuscript discussing this topic.

**5) Closing statement, line 191-194. Changes in the polar vortex shown in this paper have been limited to the region above about the 10 mbar level. No connections from those levels to the tropospheric annular mode have been established by this paper or by other work, and I don't think it's appropriate to speculate about them here. The paper's conclusions section should reflect only what it has actually concluded and avoid speculation of this type.**

We rephrased this part as: "Seasonal stratospheric ozone depletion due to the descending indirect NOx has been also shown to influence stratospheric temperatures and the polar vortex (Arsenovic et al., 2016; Salminen et al., 2019; Asikainen et al., 2020). Thus, there is a great potential of improving future projections and seasonal variability of the polar stratosphere by implementing a more accurate solar forcing, including EEP to the earth system models (Matthes et al., 2017).

---

## Author Response (AR3)

**Reply to Editor**

We have added your comments below in **bold blue font** and give our replies in black.

**I am happy to accept your paper to ACP.**
**I think it should be "net chemical ozone production" in line 22 and "enhanced net ozone production" in line 174 but I leave that up to you, if you think this correction is needed.**

Suggested phrases are now used.